Medical Imaging with Deep Learning 2023

# Deep Learning-Based Segmentation of Locally Advanced Breast Cancer on MRI in Relation to Residual Cancer Burden: A Multi-Institutional Cohort Study

**Mark Janse** [1]                                    M.H.A.JANSE-2@UMCUTRECHT.NL
**Liselore Janssen** [1]                              L.M.JANSSEN-11@UMCUTRECHT.NL
**Bas van der Velden** [1]                            B.H.M.VANDERVELDEN-2@UMCUTRECHT.NL
**Maaike Moman** [1,2]                                MAAIKEMOMAN@ALEXANDERMONRO.NL
**Elian Wolters-van der Ben** [3]                     E.WOLTERS@ANTONIUSZIEKENHUIS.NL
**Marc Kock** [4]                                     KOCKM@ASZ.NL
**Max Viergever** [1]                                 M.A.VIERGEVER-2@UMCUTRECHT.NL
**Paul van Diest** [1]                                P.J.VANDIEST@UMCUTRECHT.NL
**Kenneth Gilhuijs** [1]                              K.G.A.GILHUIJS@UMCUTRECHT.NL

[1] *University Medical Center Utrecht, Utrecht, The Netherlands*

[2] *Alexander Monro hospital, Bilthoven, The Netherlands*

[3] *St. Antonius hospital, Nieuwegein, The Netherlands*

[4] *Albert Schweitzer hospital, Dordrecht, The Netherlands*

This paper was previously published as (Janse et al., 2023)

## Abstract

While several methods have been proposed for automated assessment of breast-cancer response to neoadjuvant chemotherapy on breast MRI, limited information is available about their performance across multiple institutions. In this paper, we assess the value and robustness of nnU-Net-derived volumes of locally advanced breast cancer (LABC) on MRI to infer the presence of residual disease after neoadjuvant chemotherapy. An nnU-Net was trained to segment LABC on a single-institution training set and validated on a multi-center independent testing cohort. Based on resulting tumor volumes, an extremely randomized tree model was trained to assess residual cancer burden (RCB)-0/I vs. RCB-II/III. An independent model was developed using functional tumor volume (FTV). Models were tested on an independent testing cohort, response assessment performance and robustness across multiple institutions were assessed. Results show that nnU-Net accurately estimate changes in tumor load on DCE-MRI, that these changes associated with RCB after NAC, and that they are robust against variations between institutions.

**Keywords:** Breast MRI, segmentation, deep learning, response monitoring, locally advanced breast cancer

## 1. Introduction

Neoadjuvant chemotherapy (NAC) is increasingly used to treat patients with breast cancer as it allows monitoring of treatment response with the tumor *in situ* thus offers opportunity for more personalized treatment. The most sensitive modality to visualize tumor extent in three dimensions is dynamic contrast-enhanced magnetic resonance imaging (DCE-MRI).

Methods for response monitoring on MRI range from manual assessment by radiologists to methods being investigated for fully automated analysis. Manual assessment has been shown to be predictive of pathological complete response (pCR), depending on tumor subtype (Janssen et al., 2022). Combinations of manually selecting regions of interest (ROI) and

semi-automatic thresholding have also been proposed, including a semi-automated method to establish functional tumor volume (FTV) (Hylton et al., 2016). Fully automated data-driven methods to assess response to NAC on MRI have also been proposed, using radiomics or deep learning (Choi et al., 2019; Comes et al., 2021; Joo et al., 2021). Little is known about the robustness of these methods across multiple institutions.

This study aimed to establish whether nnU-Net accurately assesses changes in tumor load on DCE-MRI that are associated with residual cancer burden (RCB) after NAC robust to variations between institutions and MRI scanners. Secondly, whether such model is in agreement with the relationship between FTV and RCB.

## 2. Methods

### 2.1. Datasets

The training cohort consisted of 105 consecutively included breast cancer cases treated with neoadjuvant chemotherapy (NAC) in a single institution. Patients underwent two MRI examinations: The first at baseline prior to NAC, and the second either midway through the chemotherapy schedule, or immediately before the second-to-last cycle of chemotherapy. The second MRI examination was defined as the follow-up examination. Either 1.5 T or 3 T Philips scanners were used in this set. The independent testing cohort consisted of 54 consecutively included breast cancer cases treated with NAC in four institutions. The baseline scan before any treatment were used as well as the follow-up scans after all cycles of NAC but prior to surgery. Imaging in the testing cohort was performed exclusively on 3 T scanners, both Philips (Hospital 1 and 4) and Siemens (Hospital 2 and 3). All examinations had a pre-contrast scan and up to five post-contrast scans acquired at intervals of 55 to 89 seconds, fat-supressed was used. Ground-truth annotations of all training scans were derived from a previously reported histopathology-validated semi-automated region grower (Alderliesten et al., 2007).

To evaluate the response to NAC on histopathology, the residual cancer burden RCB was derived from the final post-surgery resection specimens following the methodology described by Symmans *et al* (Symmans et al., 2007). The RCB score was dichotomized into two categories: RCB categories RCB-0 (i.e. pCR, pathological complete response) and RCB-I were defined as good responders to NAC. Conversely, categories RCB-II and RCB-III were considered bad responders.

### 2.2. Response assessment

The ground truth segmentations were used to train a 3D nnU-Net CNN (Isensee et al., 2021). Input to the nnU-Net were the precontrast DCE series and five postcontrast DCE MRI (i.e. six channels in total). To evaluate segmentation performance, two-fold cross-validation was performed on the training set, while for final evaluation of response assessment performance, the network was trained on the entire training set.

To compare nnU-Net to a previously validated method for response assessment, Functional Tumor Volume (FTV) was also calculated per breast from each MRI examination following the description by Newitt *et al.* (Newitt et al., 2014).

Table 1: Performance of tumor response assessment in terms of residual cancer burden (RCB) on a per-hospital basis for the nnU-Net and FTV segmentation methods. Numbers presented are areas under the receiver operator curve (AUC).

|                       | MR vendor | nnU-Net | FTV  |
| --------------------- | --------- | ------- | ---- |
| Hospital 1 ($n = 12$) | Philips   | 0.63    | 0.71 |
| Hospital 2 ($n = 21$) | Siemens   | 0.74    | 0.75 |
| Hospital 3 ($n = 19$) | Siemens   | 0.79    | 0.81 |
| Hospital 4 ($n = 2$)  | Philips   | 1.00    | 1.00 |

An extremely randomized tree model was fit to assess tumor response to NAC (Geurts et al., 2006). Three input candidate predictors were used: the lesion volume determined on the follow-up scan, tumor subtype (HER2+, HER2-/ER+ or triple negative) and the difference in tumor volume between baseline and follow-up. The end point was tumor response expressed as the dichotomized RCB. The area under the receiver operator curve (AUC) was used as measure of model performance. Five-fold nested cross-validation was performed for hyperparameter tuning and internal model validation. Two separate models were trained, one where the volumes were determined using the previously trained nnU-Net, the second one using FTV.

## 3. Results and conclusion

The median (interquartile range (IQR)) cross-validated Dice score from the nnU-Net, was 0.87 (0.62-0.93). Pearson's correlation between volumes derived from the nnU-Net and the ground truth was R=0.95 (fold 1: $R = 0.93$, fold 2: $R = 0.97$). The correlation between the nnU-Net-derived volume and FTV in the training cohort was $R = 0.74$ for the baseline scan, $R = 0.72$ for follow-up, and $R = 0.80$ for all scans combined. All correlations were statistically significant ($P < 0.05$). In the testing cohort, the median (IQR) AUC of the response assessment model was 0.76 (0.71-0.84) for nnU-Net-derived tumor volumes and 0.77 (0.74-0.86) for FTV. There was no significant difference in AUC between the two models ($p = 0.66$). Per hospital performance varied, with the worst performance associated with the hospital from the training set (Hospital 1) (Table 1).

We conclude that nnU-Net can accurately estimate changes in tumor load on DCE-MRI and that these changes are associated with RCB after NAC. The response assessment is on par with that derived using FTV, a previously validated method, but it is fully automated and therefore observer independent. The performance of the model appears to be robust to variations in scan parameters across multiple institutions, proving the versatility of the method.

## Acknowledgments

The authors would like to thank R. Offenberg for her help in preparing and analyzing the data. This research was funded by the European Union Horizon 2020 research and innovation program under grant agreement no. 755333 (LIMA).

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
