# OpenReview forum: "Deep Learning-Based Segmentation of Locally Advanced Breast Cancer on MRI in Relation to Residual Cancer Burden: A Multi-Institutional Cohort Study"
_MIDL.io/2023/Short_Paper_Track — MIDL 2023 Short paper track Poster_

### Official Review · Reviewer_dwbF · 2023-04-20
**More testing cases/institutions, unclear model generalizability, and lacking comparisons to other segmentation models**

**Rating:** 5
**Confidence:** 4

**Review:**

The paper presents an evaluation of the nnU-Net model for detecting residual disease in locally advanced breast cancer patients after neoadjuvant chemotherapy. The study used multi-center independent testing and an extremely randomized tree model to assess residual cancer burden, and the results demonstrate the nnU-Net model's ability to accurately estimate changes in tumor load on DCE-MRI. However, to enhance the study's impact, future work should consider increasing the number of independent testing cases and institutions. It is also important to note the potential limitations of the study, such as the unclear generalizability of the segmentation model trained on images from Philips scanners to images from GE and Siemens scanners. Moreover, a comparison with other segmentation models, such as the transformer U-Net model, would provide valuable insights into the performance of nnU-Net in comparison to other state-of-the-art methods.

---

### Official Review · Reviewer_G2TA · 2023-04-23
**This paper evaluates the performance and robustness of nnU-Net, a deep learning model, in estimating changes in tumor load on breast MRI for assessing residual cancer burden after neoadjuvant chemotherapy. The study shows that nnU-Net is accurate in estimating tumor changes, associated with residual cancer burden, and robust against variations between institutions, and its performance is in agreement with functional tumor volume assessment.**

**Rating:** 6
**Confidence:** 4

**Review:**

In this paper, the authors trained an nnU-Net to segment locally advanced breast cancer (LABC) on MRI and used it to estimate tumor volume changes associated with residual cancer burden (RCB) after neoadjuvant chemotherapy. They found that nnU-Net accurately estimated changes in tumor load on DCE-MRI and that the method was robust against variations between institutions and MRI scanners.

**Pros:**

- The paper is clearly written and the experiments are sound.
- The authors attempted to use nnU-Net to measure changes in breast tumor load.
- The authors aimed to test the generalizability of this automated approach on test DCE-MRI data from multiple institutions.
- The AUC results show consistent performance compared to the standard semi-automated functional tumor volume (FTV) approach, which requires selecting a region-of-interest.
- The authors conducted statistical tests to quantify the correlation between nnU-Net predicted volume and FTV.

**Cons:**

- The authors missed providing some qualitative results for both nnU-Net and FTV, so the reader can understand the challenge in this task and also appreciate the performance improvement compared to the conventional approach.
- It would be great if the authors reported the improvements in terms of time between nnU-Net and FTV. How much could it reduce the breast assessment time compared to the semi-automated approach?
- In Table 1, the authors should discuss the reasons for performance discrepancy between hospitals. For example, what could be the reasons that nnU-Net didn't perform well on Hospital 1 compared to FTV?
- The training set and test cohort hospital have a very small sample size. It would be difficult to test the generalizability of this approach using only two cases from a hospital (e.g. hospital 4) to appreciate. Also,
- Some details are missing regarding how they trained the nnU-Net model using DCE-MRI images. Did they train on whole images or only the breast area?
- Are the data from 4 external hospitals also acquired using Philips scanners (2T, 1.5T)? or are they from different vendors like GE, Siemens?
- A comparison with other state-of-the-art deep learning methods for locally advanced breast cancer is missing.